# A Player Selection Network
# for Scalable Game-Theoretic Prediction and Planning

**Tianyu Qiu**                                          *tianyuqiu@utexas.edu*
*University of Texas at Austin*
*Austin, United States of America*

**Eric Ouano**                                          *eouano@utexas.edu*
*University of Texas at Austin*
*Austin, United States of America*

**Fernando Palafox**                                    *fernandopalafox@utexas.edu*
*University of Texas at Austin*
*Austin, United States of America*

**Christian Ellis**                                     *christian.ellis@austin.utexas.edu*
*University of Texas at Austin*
*Austin, United States of America*

**David Fridovich-Keil**                                *dfk@utexas.edu*
*University of Texas at Austin*
*Austin, United States of America*

**Reviewed on OpenReview:** *https://openreview.net/forum?id=YvvB78ILSP*

## Abstract

While game-theoretic planning frameworks are effective at modeling multi-agent interactions, they require solving large optimization problems where the number of variables increases with the number of agents, resulting in long computation times that limit their use in large-scale, real-time systems. To address this issue, we propose i) **PSN Game**—a learning-based, game-theoretic prediction and planning framework that reduces game size by learning a *Player Selection Network* (PSN); and ii) a *Goal Inference Network* (GIN) that makes it possible to use the PSN in incomplete-information games where other agents' intentions are unknown to the ego agent. A PSN outputs a player selection mask that distinguishes influential players from less relevant ones, enabling the ego player to solve a smaller, masked game involving only selected players. By reducing the number of players included in the game, PSN shrinks the corresponding optimization problems, leading to faster solve times. The PSN Game framework is more flexible than existing player selection methods as it i) relies solely on observations of players' past trajectories, without requiring full state, action, or other game-specific information; and ii) requires no online parameter tuning. Experiments in both simulated scenarios and real-world pedestrian trajectory datasets show that PSN is competitive with, and often improves upon, the evaluated explicit game-theoretic selection baselines in i) prediction accuracy and ii) planning safety. Across scenarios, PSN typically selects substantially fewer players than are present in the full game, thereby reducing game size and planning complexity. PSN also generalizes to settings in which agents' objectives are unknown, via the GIN, without test-time fine-tuning. By **selecting only the most relevant players** for decision-making, PSN Game provides a practical mechanism for **reducing planning complexity** that can be integrated into existing multi-agent planning frameworks.[1]

---

[1]Code is available at `https://github.com/CLeARoboticsLab/player-selection-game`.

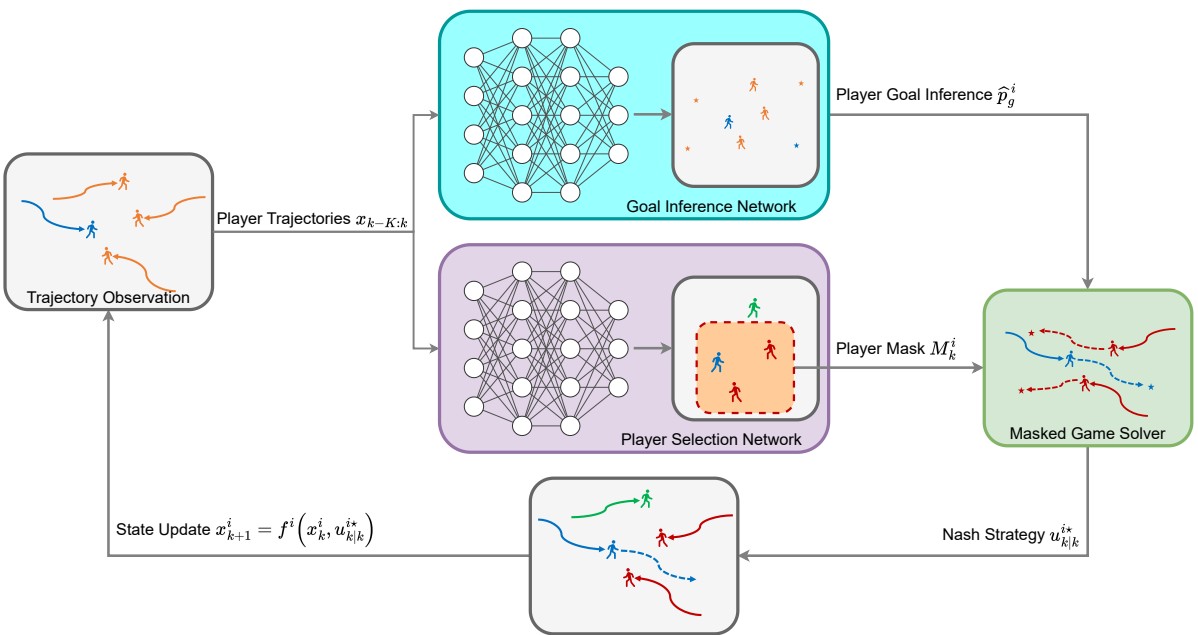

Figure 1: Overview of our game-theoretic prediction and planning framework via the Player Selection Network (PSN) and the Goal Inference Network (GIN). At each time step, the ego player (blue) observes other agents' past trajectories and inputs them to PSN. This explicit reduced-game construction supports repeated receding-horizon solves under onboard compute limits. The network selects important players (red) and excludes less relevant ones (green). The ego player then solves a masked game over the selected subset to obtain the Nash strategy $u_{k|k}^{i\star}$ and updates its state using equation 1b.

# 1 Introduction

Game-theoretic planning frameworks are widely used in robotics to model multi-agent interactions in terms of Nash (or other) equilibrium strategies, which can be identified by optimization-based algorithms (Fridovich-Keil et al., 2020; Sun, 2025). Most existing approaches, however, focus on static scenarios or involve only a small number of agents (e.g., $\leq 5$), where computational efficiency is rarely a concern. In contrast, scenarios involving many dynamic agents demand frequent replanning, making computational efficiency a critical bottleneck. In such settings, the number of optimization variables, typically proportional to the number of agents, can grow rapidly and lead to significant delays. Prior work (Laine et al., 2023; Fridovich-Keil et al., 2020) has shown that even in simple linear-quadratic games, computation time scales cubically with the total number of state and control variables for all agents, rendering game-theoretic planners impractical for large-scale use.

Similar limitations are observed in human driving behaviors. Prior research on drivers' attention (Gershon et al., 2019; Fang et al., 2021; Morando et al., 2018; Li et al., 2019; Xia et al., 2020) shows that in dense traffic, monitoring all surrounding vehicles simultaneously is difficult. Although robots are not subject to distraction, they still face onboard compute limits in repeated solver-in-the-loop planning as the number of agents increases. These observations suggest two key insights: i) it is infeasible to consider all agents at once; and ii) focusing on a strategically selected subset of agents is often sufficient for safe interaction.

For game-theoretic planning, Chahine et al. (2023) proposed ranking-based approaches that prioritize agents based on simple heuristics, such as proximity to the ego player or their impact on cost. However, these approaches face the following limitations: i) they often rely on access to control inputs or game-specific parameters, which are typically unavailable in practice; and ii) using a fixed number of selected players requires manual tuning and may either exclude important agents or include irrelevant ones.

To address these limitations, we propose PSN Game—a novel game-theoretic framework that learns a Player Selection Network (PSN) to reduce game size in multi-agent trajectory prediction and planning. The PSN takes the ego agent and surrounding agents' past trajectories as input and outputs a player-selection mask

identifying the most influential agents. Furthermore, we build a Goal Inference Network (GIN) that infers other agents' objectives even when they are unknown *a priori*, making player selection practical even in incomplete-information settings. Ultimately, the PSN, together with the GIN, constructs a smaller-scale masked game over the selected subset of agents, whose equilibrium solution can encode trajectory prediction and planning interactions (Fig. 1). Our contributions are fourfold:

i. We introduce an objective-driven Player Selection Network (PSN), trained with a differentiable dynamic game solver that enables backpropagation through the selection mask.

ii. We introduce a supervised Goal Inference Network (GIN) that infers agents' goals from past trajectories, allowing the PSN to operate in scenarios where agents' intentions are unknown.

iii. We develop a receding-horizon game-theoretic planning framework that utilizes the PSN to identify equilibrium strategies efficiently by solving reduced-size games.

iv. We empirically validate PSN Game in multi-agent simulations and real-world pedestrian trajectory datasets. PSN Game typically reduces the game size by 50% to 75% while maintaining strong prediction and planning performance relative to full-player games and the evaluated baselines.

PSN Game offers two key advantages over prior selection methods: i) Flexible data usage: at runtime, the PSN relies only on past trajectory data—either full-state (e.g., position and velocity) or partial-state (e.g., position only)—without requiring control inputs, cost functions, or other game-specific parameters. It also eliminates the need for online parameter tuning. ii) Strong empirical performance: by leveraging spatio-temporal patterns in past trajectories, PSN achieves strong and consistent performance in prediction and planning metrics relative to the evaluated explicit selection baselines. These features make PSN Game broadly applicable across diverse multi-agent planning and prediction scenarios.

## 2 Related Work

### 2.1 Game-theoretic Planning and Time Complexity Analysis

Game-theoretic planning has been widely applied to model diverse multi-agent scenarios, including intention inference (Le Cleac'h et al., 2021; Mehr et al., 2023; Peters et al., 2023; Liu et al., 2024b), hierarchical interactions (Khan & Fridovich-Keil, 2024; Hu et al., 2024; Khan et al., 2026), handling occluded agents in autonomous driving (Zhang & Fisac, 2021; Gupta & Fridovich-Keil, 2023; Qiu & Fridovich-Keil, 2024), and robot formation tasks (Liu et al., 2024a; Qiu et al., 2026). These methods typically consider small-scale environments with relatively few agents, where real-time equilibrium computation using solvers such as (Dirkse & Ferris, 1995; Fridovich-Keil et al., 2020; Laine et al., 2023) is feasible. These Newton-style solvers work by iteratively approximating the underlying noncooperative game with a sequence of quadratic problems that can be solved analytically; however, the complexity of each subproblem scales cubically with the number of players, as illustrated in Fig. 2, which makes it nontrivial to solve larger-scale problems involving many agents.

To the best of our knowledge, absent any additional structure, e.g., homogeneity of agents' objectives that admits a mean field representation (Lasry & Lions, 2007), this complexity presents a fundamental obstacle for deploying game-theoretic planning in large-scale, real-time multi-agent systems. Given that the horizon $T$ and state dimension $n$ for each agent are often fixed by the task, reducing the number of agents in the game remains the most viable strategy for improving computational performance.

### 2.2 Player Selection in Multi-agent Interactions

Existing methods for selecting important players in multi-agent settings generally fall into two categories: threshold-based and ranking-based approaches.

**Threshold-based methods** highlight agents that breach a predefined distance threshold around the ego agent. This idea has been widely adopted in works such as (Trautman & Krause, 2010; Chen et al., 2017),

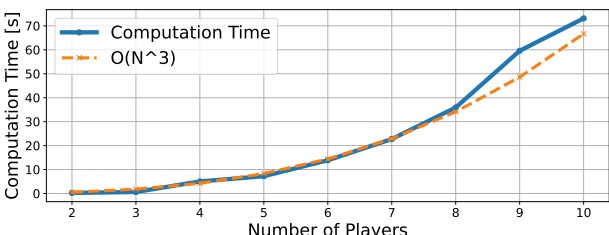

Figure 2: Computation time from (Fridovich-Keil) versus the number of players in the game given in Section 4.1.1. Time consumption grows cubically as player number increases.

where a safety margin is first defined to select key agents before applying prediction or planning frameworks. However, these methods heavily rely on threshold tuning, making them hard to generalize across scenarios.

**Ranking-based methods**, by contrast, select the top-$k$ most influential agents according to a heuristic, often by distance to the ego agent; common examples include nearest-neighbor selection (Van den Berg et al., 2008; Snape et al., 2011). In learning-based approaches (Everett et al., 2018; Chen et al., 2019; Wang et al., 2025), pooling mechanisms are employed to fix the input size to a neural network trained to identify nearest neighbors. In the context of game-theoretic planning, Chahine et al. (2023); Vakil & Pierson (2026) introduced ranking metrics based on, e.g., the sensitivity of the ego agent's cost to other agents' state.

However, ranking-based methods present several challenges. Selecting a fixed number of agents can be too aggressive (omitting critical players) or too conservative (including irrelevant ones), often requiring environment-specific parameter tuning. Furthermore, many methods assume access to agents' controls or other game-specific information, which may not be available in practice.

In contrast, our proposed PSN Game addresses the limitations of both threshold-based and ranking-based approaches. PSN Game is lightweight in information requirements at runtime. PSN-Partial uses only past position observations, while PSN-Full uses past state trajectories; neither variant requires controls, cost functions, or other game-specific information at test time. In particular, although cost functions are used during training to generate trajectories, PSN Game does not require that information at test time, enabling flexible deployment across different environments with model mismatch. Experiments show that PSN remains competitive on safety and often produces smoother, shorter ego trajectories relative to the evaluated baselines across a range of multi-agent navigation scenarios.

# 3 Learning to Identify Key Players in Noncooperative Interactions

## 3.1 Preliminaries: Nash Games

A finite horizon, discrete-time, open-loop Nash game with $N$ agents is characterized by their respective states $x_k^i \in \mathbb{R}^n$ and control inputs $u_k^i \in \mathbb{R}^m$, $i \in \{1, \dots, N\} \equiv [N]$, over time $k \in [T]$, where $T$ is the planning horizon. Each agent $i$'s state transition from time $k$ to $k+1$ is governed by the dynamics $x_{k+1}^i = f^i(x_k^i, u_k^i)$. We denote agent $i$'s trajectory and control sequence compactly as $\mathbf{x}^i := (x_{0:T}^i)$ and $\mathbf{u}^i := (u_{0:T-1}^i)$; and similarly, all agents' states/control inputs at time $k$ by $\mathbf{x}_k := (x_k^{1:N})$, $\mathbf{u}_k := (u_k^{1:N})$ and their trajectory/control sequence over time by $\mathbf{x} := (\mathbf{x}_{0:T})$, $\mathbf{u} := (\mathbf{u}_{0:T-1})$, respectively. The function $J^i(\mathbf{x}, \mathbf{u}; \theta^i) := \sum_{k=0}^T c_k^i(\mathbf{x}_k, \mathbf{u}_k; \theta^i)$ defines the $i^{\text{th}}$ agent's cumulative cost, with game-specific parameter $\theta^i$. The game is thus fully characterized by all agents' costs parameterized by $\theta := (\theta^{1:N})$, initial states $\mathbf{x}_0$, and dynamics $\mathbf{f} := (f^{1:N})$, and is denoted as $\Gamma(\mathbf{x}_0, \mathbf{f}; \theta)$. In this game, each agent aims to minimize its cost while adhering to dynamics, i.e., agent $i$ solves the optimization problem minimizing the time-cumulative sum of private cost $c_{p,k}^i$ and shared cost

$c_{s,k}^i = \sum_{\substack{j=1 \\ j \neq i}}^{N} c_k^{ij}$, where $c_k^{ij}$ is the mutual shared cost:

$$\min_{\mathbf{x}^i, \mathbf{u}^i} \quad \underbrace{\sum_{k=0}^{T} [c_{p,k}^i(x_k^i, u_k^i; \theta^i) + c_{s,k}^i(\mathbf{x}_k, \mathbf{u}_k; \theta^i)]}_{J^i(\mathbf{x}, \mathbf{u}; \theta^i) = \sum_{k=0}^{T} c_k^i(\mathbf{x}_k, \mathbf{u}_k; \theta^i)} \tag{1a}$$

$$\text{s.t.} \quad x_{k+1}^i = f^i(x_k^i, u_k^i), \quad k \in [T], \tag{1b}$$

so each agent's optimization problem is inherently coupled with the problems of the other agents.

**Open-Loop Nash Equilibrium**: If inequalities

$$J^i(\mathbf{x}^*, \mathbf{u}^*; \theta^i) \leq J^i(\mathbf{x}^i, \mathbf{x}^{-i*}, \mathbf{u}^i, \mathbf{u}^{-i*}; \theta^i), \quad i \in [N], \tag{2}$$

concerning state and control trajectories of agent $i$ $(\mathbf{x}^i, \mathbf{u}^i)$ and other agents $(\mathbf{x}^{-i}, \mathbf{u}^{-i})$ are satisfied for all $\mathbf{x}^i, \mathbf{u}^i$ that remain feasible with respect to equation 1b, then $\mathbf{u}^{i*}$ is called an Open-Loop Nash equilibrium (OLNE) strategy with $\mathbf{x}^{i*}$ being the corresponding OLNE state trajectory for agent $i$. This inequality indicates that no agent can reduce their cost by unilaterally deviating from $\mathbf{u}^{i*}$ (Başar & Olsder, 1998).

*Differentiability of Nash game solvers* A key property of solutions $(\mathbf{x}^*, \mathbf{u}^*)$ to generalized Nash problems is that they are (directionally) differentiable with respect to problem parameters $\theta$, as discussed in (Liu et al., 2023). In our work, we employ the solver in (Sun, 2025), which leverages implicit differentiation to efficiently compute those derivatives and thus enables backpropagation during training.

### 3.2 Masked Nash Game with Selected Players

We first define the concept of a *player selection mask*, which helps the ego agent to select the important agents during planning:

**Definition 1** (Player Selection Mask). Suppose that the $i^{\text{th}}$ player is the ego agent. Then, the player selection mask is denoted as $M^i := (m^{ij}) \in \{0, 1\}^{N-1}$ for $j \in [N] \setminus \{i\}$, where

$$m^{ij} := \begin{cases} 1, & \text{Agent } j \in [N] \text{ is included in the game} \\ 0, & \text{Agent } j \in [N] \text{ is excluded from the game.} \end{cases} \tag{3}$$

Given the player selection mask $M^i$, we construct the *Masked Nash Game* as follows:

**Definition 2** (Masked Nash Game). Given an open-loop Nash game $\Gamma(\mathbf{x}_0, \mathbf{f}; \theta)$, a masked Nash game for agent $i$ is denoted as $\Gamma^i(\tilde{\mathbf{x}}_0, \tilde{\mathbf{f}}; \tilde{\theta})$, where we define the set of unmasked agents (agents that are included in the game) $U^i := \{j \neq i \mid m^{ij} = 1\}$ and let

$$\tilde{\theta} := (\theta^j)_{j \in U^i}, \quad \tilde{\mathbf{x}}_k := (x_k^j)_{j \in U^i}, \quad \tilde{\mathbf{f}} := (f^j)_{j \in U^i}, \quad \tilde{c}_{s,k}^i(\mathbf{x}_k, \mathbf{u}_k; \theta^i) := \sum_{j \in U^i} c_{s,k}^{ij}(\mathbf{x}_k, \mathbf{u}_k; \theta^i). \tag{4}$$

The masked game retains only the states, dynamics, and parameters associated to those agents most relevant to agent $i$. Agent $i$ then solves the masked Nash game $\Gamma^i(\tilde{\mathbf{x}}_0, \tilde{\mathbf{f}}; \tilde{\theta}, M^i)$, and employs its masked OLNE strategy $\tilde{\mathbf{u}}^{i*}$ as its own plan.

### 3.3 Player Selection Network (PSN)

Given this construction, we now turn to the challenge of identifying the set of key players that ego agent $i$ should consider in its masked game model of the interaction in order to maintain safety and efficiency while minimizing the computational burden of solving the corresponding game to obtain its strategy at each time $k$. We propose the Player Selection Network (PSN), which is a neural network whose task is to infer a mask $M^i$ that balances performance with sparsity. Based on the available information, we present two variants of

the network: PSN-Full, whose input is all agents' past states $\mathbf{x}_{0:K}$, and PSN-Partial[2], whose input is only a partial observation of state, i.e. $\{h(\mathbf{x}_k)\}_{k=0}^{K}$ for a known function $h$. For example, $h(\mathbf{x}_k)$ could return only the Cartesian position of all agents on the road, and not their velocities. Both variants' output is the inferred mask $M^i$. Player $i$ will then solve the masked game $\Gamma^i(\tilde{\mathbf{x}}_0, \tilde{\mathbf{f}}; \tilde{\theta}, M^i)$ to obtain its strategy $\tilde{\mathbf{u}}_{K:K+T}^i$.

**Remark 1.** *In the ideal case, the mask $m^{ij}$ should be binary as defined in equation 3, which hinders backpropagation and the combinatorial mask space contains $2^{N-1}$ possible masks $M^i$. Although it is possible to train a neural network with categorical outputs via the Gumbel softmax reparameterization (Jang et al., 2017), it becomes expensive when $N$ is large. To overcome such limitations, we set each mask entry to a continuous value $m^{ij} \in [0, 1]$, and relax the shared cost $c_{s,k}^i(\boldsymbol{x}_k, \boldsymbol{u}_k; \theta^i)$ as $\hat{c}_{s,k}^i(\boldsymbol{x}_k, \boldsymbol{u}_k; \theta^i)$, where*

$$\hat{c}_{s,k}^i(\boldsymbol{x}_k, \boldsymbol{u}_k; \theta^i) \coloneqq \sum_{\substack{j=1 \\ j \neq i}}^{N} m^{ij} c_k^{ij}(\boldsymbol{x}_k, \boldsymbol{u}_k; \theta^i). \tag{5}$$

*This yields a soft mask that continuously reweights shared interaction costs. At runtime, a threshold $m_{th}$ converts the continuous PSN output to binary masks (1 if larger than $m_{th}$ and 0 otherwise).*

**Loss Function Design**: We train two PSN variants: PSN-Prediction and PSN-Planning for prediction and planning tasks, respectively; their corresponding training loss functions are structured as:

$$L_{\text{Pred}} = L_{\text{Binary}} + \sigma_1 L_{\text{Sparsity}} + \sigma_2 L_{\text{Similarity}}, \quad L_{\text{Plan}} = L_{\text{Binary}} + \sigma_3 L_{\text{Sparsity}} + \sigma_4 L_{\text{Cost}} \tag{6}$$

$$L_{\text{Binary}} = \frac{1}{N} \sum_{j=1}^{N} m^{ij}(1 - m^{ij}), \ L_{\text{Sparsity}} = \frac{\|M^i\|_1}{N}, \ L_{\text{Similarity}} = \sum_k \|h(\hat{\mathbf{x}}_k^i) - p_k^i\|_2, \ L_{\text{Cost}} = J^i(\hat{\mathbf{x}}^i; \mathbf{x}^{\neg i}).$$

with respective nonnegative weights $\sigma_{1:4}$. Here, $p_k^i$ denotes the observed position of ego agent $i$ at time $k$, and $\mathbf{x}^{\neg i}$ is the other agents' trajectory if ego agent $i$ were to consider all other agents; in contrast, $\hat{\mathbf{x}}^i$ is agent $i$'s computed Nash trajectory for the game with relaxed shared cost in equation 5 (i.e., returned by the aforementioned differentiable game solver). $L_{\text{Binary}}$ encourages mask $m^{ij}$ to converge to either 0 or 1. $L_{\text{Sparsity}}$ encourages agent $i$ to consider fewer other agents. $L_{\text{Similarity}}$ encourages agent $i$ to consider agent subsets which lead to Nash solutions that recover its observed trajectory, and $L_{\text{Cost}}$ encourages agent $i$ to consider appropriate agents to minimize its game cost.

### 3.4 Goal Inference Network (GIN)

In order to employ the PSN in more complicated incomplete-information scenarios (i.e., where the parameter $\theta$ introduced in Section 3.1 is unknown), it is necessary to recover $\theta$ from observations of agents' states. In this work, we focus on treating $\theta^i$ as agent $i$'s 2-D goal position $\hat{p}_g^i$. More generally, $\theta^i$ can represent other parameters, such as weights on basis functions spanning agent preferences (Peters et al., 2021; Liu et al., 2023; Qiu & Fridovich-Keil, 2024). We introduce a data-driven goal inference network $G_\phi$, which infers an agent's goal $\hat{p}_g^i$ from partial observations of that agent's past trajectory $\{h(\mathbf{x}_k)\}_{k=0}^{K}$. The network is trained from the dataset $\mathcal{D}$, which contains the Nash equilibrium trajectories of all agents from solving a game with ground truth goals, by minimizing the mean goal inference error $L_{\text{Goal}}$ for each sample $d$:

$$\min_\phi \underbrace{\frac{1}{|\mathcal{D}| \cdot N} \sum_{d \in \mathcal{D}} \sum_{i \in [N]} \|p_{g,\text{ref}}^i - G_\phi(\mathbf{x}_{0:K})\|}_{L_{\text{Goal}}} \tag{7}$$

Assisted by $G_\phi$, the PSN is readily adaptable to trajectory prediction and ego-centric planning tasks where the agents' goals are not known to the predictor/planner, by constructing and solving a masked game $\Gamma^i(\tilde{\mathbf{x}}_k, \tilde{\mathbf{f}}; \{\hat{p}_g^i\}_{i=1}^{N}, M_k^i)$ parametrized by the inferred goals.

---

[2]While PSN-Partial is capable of selecting key agents from partial observations, we emphasize that solving the masked game still requires the knowledge of all agents' full states. Prior works (Peters et al., 2021; Qiu & Fridovich-Keil, 2024) investigate methods to estimate full states of agents from partial observations via inverse games. PSN-Partial is designed to integrate with methods where only partial states of agents are needed.

---

**Algorithm 1** Receding Horizon Prediction via PSN

---

**Require:** PSN-Prediction (Full, Partial), Goal Inference Network $G_\phi$, receding horizon masked game $\Gamma^i(\tilde{\mathbf{x}}_k, \tilde{\mathbf{f}}; \{p_g^i\}_{i=1}^N, M_k^i)$, observation interval $K$, prediction interval $T_{\text{Pred}}$

**Ensure:** Agent $i$'s receding horizon trajectory prediction $\hat{\mathbf{x}}_{K+1:K+T_{\text{Pred}}}^i$

1: $\{\hat{p}_g^i\}_{i=1}^N \leftarrow G_\phi(\{h(\mathbf{x}_k)\}_{k=0}^K)$.          $\triangleright$ Infer agents' goals via $G_\phi$ if $\{p_g^i\}_{i=1}^N$ is not known

2: **for** $k = K, \ldots, K + T_{\text{Pred}} - 1$ **do**

3:      $M_k^i \leftarrow$ PSN-Full$(\hat{\mathbf{x}}_{k-K:k})$ or PSN-Partial$(\hat{\mathbf{p}}_{-K:k})$          $\triangleright$ Identify key agents via PSN

4:      $u_{k|k}^{i*} \leftarrow$ solution of $\Gamma_k^i(\tilde{\mathbf{x}}_k, \tilde{\mathbf{f}}; \{\hat{p}_g^i\}_{i=1}^N, M_k^i)$          $\triangleright$ Solve masked game for agent $i$

5:      $u_{k|k}^{-i*} \leftarrow$ solution of $\Gamma_k(\mathbf{x}_k, \mathbf{f}; \{\hat{p}_g^i\}_{i=1}^N)$          $\triangleright$ Solve full game for the other agents

6:      $\hat{x}_{k+1}^i \leftarrow f^i(x_k^i, u_{k|k}^i)$          $\triangleright$ Update ego-agent prediction via equation 1b

7:      $\hat{x}_{k+1}^{-i} \leftarrow f^{-i}(x_k^{-i}, u_{k|k}^{-i*})$          $\triangleright$ Update the other agents' predictions

8: **end for**

---

**Algorithm 2** Receding Horizon Planning via PSN

---

**Require:** PSN-Planning (Full, Partial), Goal Inference Network $G_\phi$, receding horizon masked game $\Gamma^i(\tilde{\mathbf{x}}_k, \tilde{\mathbf{f}}; \{p_g^i\}_{i=1}^N, M_k^i)$, observation interval $K$, ego agent $i$'s goal $p_g^i$

**Ensure:** Agent $i$'s receding horizon strategy $\mathbf{u}_{\text{RH}}^{i*}$

1: $\{\hat{p}_g^j\}_{\substack{j=1 \\ j \neq i}}^N \leftarrow G_\phi(\{h(\mathbf{x}_k)\}_{k=0}^K)$          $\triangleright$ Infer the other agents' goals via $G_\phi$ if $\{p_g^j\}_{\substack{j=1 \\ j \neq i}}^N$ is not known

2: **while** $k \geq K$ and player $i$ has not reached its goal $p_g^i$ **do**

3:      $M_k^i \leftarrow$ PSN-Full$(\mathbf{x}_{k-K:k})$ or PSN-Partial$(\mathbf{p}_{k-K:k})$          $\triangleright$ Identify key agents via PSN

4:      $u_{k|k}^{i*} \leftarrow$ solution of $\Gamma_k^i(\tilde{\mathbf{x}}_k, \tilde{\mathbf{f}}; \{p_g^i, \hat{p}_g^{-i}\}, M_k^i)$          $\triangleright$ Solve masked game for agent $i$

5:      $x_{k+1}^i \leftarrow f^i(x_k^i, u_{k|k}^{i*})$          $\triangleright$ Update agent $i$'s state via equation 1b

6:      $k \leftarrow k + 1$

7: **end while**

---

### 3.5 Receding Horizon Prediction and Planning

We now integrate the player selection network with a (differentiable) Nash game solver (Sun, 2025) in the receding time horizon for prediction and planning tasks.

In prediction tasks, as illustrated in Algorithm 1, the first $K$ steps of the states of all agents are observed, and their goals $\hat{p}_g$ are inferred by the GIN. At each step, PSN returns an adaptive mask $M_k^i := (m_k^{ij})$, which determines the key players for the ego agent $i$. Then an OLNE strategy is computed for the masked game, and the one-step prediction for the ego agent is obtained by applying $u_{k|k}^{i*}$. For the non-ego agents, we use actions obtained from the full game.

In practice, this complexity could be avoided by treating each non-ego player $j \neq i$ as ego and solving a masked game for that player to predict its action. To keep evaluation focused, we apply PSN only to the ego agent when reporting prediction results in Section 4.

In planning tasks, as illustrated in Algorithm 2, at each time step $k \geq K$, ego agent $i$ observes all agents' past trajectories of $K$ steps $\mathbf{x}_{k-K:k}$ and obtains an adaptive mask $M_k^i := (m_k^{ij})$ from the neural network, which determines the key players. Goals for non-ego agents are inferred once at initialization via GIN in Algorithm 2. The ego agent then solves for a GOLNE strategy $(u_{k|k}^{i*}, \ldots, u_{k+T-1|k}^{i*})$ for $\Gamma^i(\tilde{\mathbf{x}}_k, \tilde{\mathbf{f}}; \tilde{\theta}, M_k^i)$ and then implements the first control input $u_{k|k}^{i*}$ and updates its state according to equation 1b. The resulting receding-horizon strategy is denoted as $\mathbf{u}_{\text{RH}}^{i*} = (u_{k|k}^{i*}, u_{k+1|k+1}^{i*}, \ldots)$.

## 4 Experiments

In this section, we address two questions: i) how does PSN Game perform against baseline selection methods on prediction and planning tasks; and ii) can PSN Game adapt to challenging settings such as incomplete-information games (i.e., unknown objectives of non-ego agents), varying numbers of agents,

Table 1: Characteristics of player selection methods

| Category | Method | Pos. | Vel. | Ctrl. | Traj. | Game | Parameter(s) |
|----------|--------|------|------|-------|-------|------|--------------|
| Thres. | **PSN-Full (Ours)** | ✓ | ✓ | ✗ | ✓ | ✗ | - |
| | **PSN-Partial (Ours)** | ✓ | ✗ | ✗ | ✓ | ✗ | - |
| | Distance | ✓ | ✗ | ✗ | ✗ | ✗ | - |
| Ranking | **PSN-Full (Ours)** | ✓ | ✓ | ✗ | ✓ | ✗ | - |
| | **PSN-Partial (Ours)** | ✓ | ✗ | ✗ | ✓ | ✗ | - |
| | kNNs | ✓ | ✗ | ✗ | ✗ | ✗ | - |
| | Cost Evolution | ✓ | ✗ | ✗ | ✓ | Cost | - |
| | Gradient | ✓ | ✓ | ✓ | ✗ | Cost | - |
| | Hessian | ✓ | ✓ | ✓ | ✗ | Cost | - |
| | BF | ✓ | ✓ | ✗ | ✗ | Constraint | Barrier Const |
| | CBF | ✓ | ✓ | ✓ | ✗ | Constraint | Barrier Const |

and real-world pedestrian trajectory data? We train and validate PSN Game to investigate the following hypotheses:

**Hypothesis 1**: In prediction tasks, `PSN-Prediction` yields competitive or improved forecasts of the target agents' future trajectories relative to baseline selection methods.

**Hypothesis 2**: In planning tasks, `PSN-Planning` produces competitive or improved safety-efficiency tradeoffs relative to baselines (e.g., navigation/collision/control costs, smoothness, and trajectory length).

**Hypothesis 3**: PSN readily adapts to incomplete-information games when paired with the GIN.

**Hypothesis 4**: PSN scales to scenarios with more agents than seen in training, without retraining or fine-tuning.

**Hypothesis 5**: PSN transfers to real pedestrian trajectories.

## 4.1 Experiment Setup

### 4.1.1 Game Structure and PSN Training

For training and evaluation, we use the following game setup: $N$ players are moving towards their goals while avoiding each other. At each time $k$, the $i^{\text{th}}$ player's state $x_k^i \in \mathbb{R}^4$ encodes its position $p_k^i = [p_{x,k}^i, p_{y,k}^i]^\top$ and velocity $v_k^i = [v_{x,k}^i, v_{y,k}^i]^\top$, and evolves according to double-integrator dynamics[3], i.e.,

$$x_{k+1}^i = \begin{bmatrix} p_{k+1}^i \\ v_{k+1}^i \end{bmatrix} = \begin{bmatrix} I_2 & I_2\Delta t \\ \mathbf{0} & I_2 \end{bmatrix} \cdot \begin{bmatrix} p_k^i \\ v_k^i \end{bmatrix} + \begin{bmatrix} \mathbf{0} \\ I_2 \end{bmatrix} \Delta t \cdot a_k^i = \underbrace{Ax_k^i + B\Delta t u_k^i}_{f^i(x_k^i, u_k^i)}, \ k \in [T], \ i \in [N]. \tag{8}$$

where the control input $u_k^i = a_k^i = [a_{x,k}^i, a_{y,k}^i]^\top$ denotes the player's acceleration and $\Delta t$ denotes the time interval between two adjacent steps, and $I_2$ is the identity matrix in $\mathbb{R}^2$.

Adhering to dynamics constraints, the game objective is to minimize the cumulative running cost $c_k^i$ over time, where $c_k^i$ is characterized by private costs $c_{p,k}^i$ and shared costs $c_{s,k}^i$ with non-negative weighting parameters $\mathbf{w}^i = (w_{1:4}^i)$, i.e.

$$c_k^i = c_{p,k}^i + c_{s,k}^i, \ c_{p,k}^i = w_1^i \|p_k^i - p_{\text{ref},k}^i\|_2^2 + w_2^i \|v_k^i\|_2^2 + w_3^i \|u_k^i\|_2^2, \ c_{s,k}^i = \sum_{\substack{j=1 \\ j \neq i}}^N c_k^{ij} = w_4^i \sum_{\substack{j=1 \\ j \neq i}}^N \exp\left(-\|p_k^i - p_k^j\|_2^2\right).$$

This form of objective guides the $i^{\text{th}}$ player towards its destination $p_g^i$ along a goal-oriented reference path $\{p_{\text{ref},k}^i\}_{t=1}^T$, where $p_{\text{ref},k}^i = (1 - \frac{k}{T})p_0^i + \frac{k}{T}p_g^i$, with minimal velocity ($\|v_k^i\|_2^2$) and energy expenditure ($\|u_k^i\|_2^2$) while maintaining a safe distance from the others ($\exp\left(-\|p_k^i - p_k^j\|_2^2\right)$).

---

[3]We use double-integrator dynamics as a controlled benchmark that isolates the player-selection question while keeping repeated game solves tractable during training and evaluation. Our goal here is not to claim that PSN is specific to this dynamics class, but to compare selection methods under a common game model. Nonlinear dynamics can be readily solved with (Sun, 2025).

Table 2: Bootstrapped mean prediction performance in 4, 10, and 20 agent scenarios, and the CITR pedestrian dataset, with ground truth goals

| 4 Agents | Parameter | ADE [m] ↓ ($\pm 0.0147$)* | FDE [m] ↓ ($\pm 0.0257$) | Consistency ↑ ($\pm 0.0035$) | Num.P ($\pm 0.06$) | Solving time [s] ($\pm 0.009$) |
|---|---|---|---|---|---|---|
| **PSN-Full** | $m_{\text{th}} = 0.5$ | **0.1834** | 0.2785 | 0.9901 | 1.66 | 0.054 |
| **PSN-Partial** | | 0.1876 | 0.2745 | 0.9856 | 1.87 | 0.060 |
| Distance | $d_{\text{th}} = 1\text{m}$ | 0.2040 | 0.2985 | **0.9949** | 1.20 | 0.042 |
| **PSN-Full** | | 0.1839 | 0.2749 | 0.9901 | | |
| **PSN-Partial** | | **0.1816** | **0.2674** | 0.9856 | | |
| kNNs | | 0.1884 | 0.2802 | 0.9887 | | |
| Cost Evolution | $\lvert U^i \rvert = 1$ | 0.1861 | **0.2661** | 0.9800 | 2 | 0.077 |
| Gradient | | 0.1925 | 0.2817 | 0.9885 | | |
| Hessian | | 0.1938 | 0.2851 | 0.9915 | | |
| BF | | 0.1899 | 0.2812 | 0.9917 | | |
| CBF | | 0.1921 | 0.2843 | **0.9931** | | |
| **10 Agents** | Parameter | ($\pm 0.0161$) | ($\pm 0.0292$) | ($\pm 0.0014$) | ($\pm 0.11$) | ($\pm 0.011$) |
| **PSN-Full** | $m_{\text{th}} = 0.5$ | 0.2499 | 0.3610 | **0.9936** | 2.06 | 0.080 |
| **PSN-Partial** | | **0.2296** | 0.3295 | 0.9910 | 2.62 | 0.088 |
| Distance | $d_{\text{th}} = 1.5\text{m}$ | 0.2438 | 0.3638 | 0.9931 | 2.29 | 0.085 |
| **PSN-Full** | | 0.2314 | 0.3401 | **0.9936** | | |
| **PSN-Partial** | | **0.2213** | **0.3300** | 0.9909 | | |
| kNNs | | 0.2343 | 0.3485 | 0.9908 | | |
| Cost Evolution | $\lvert U^i \rvert = 2$ | 0.2339 | 0.3426 | 0.9776 | 3 | 0.096 |
| Gradient | | 0.2392 | 0.3517 | 0.9930 | | |
| Hessian | | 0.2360 | 0.3489 | 0.9928 | | |
| BF | | 0.2369 | 0.3508 | 0.9923 | | |
| CBF | | 0.2368 | 0.3519 | 0.9933 | | |
| **20 Agents** | Parameter | ($\pm 0.0245$) | ($\pm 0.0411$) | ($\pm 0.0023$) | ($\pm 0.11$) | ($\pm 0.011$) |
| **PSN-Full** | $m_{\text{th}} = 0.5$ | 0.3270 | 0.4823 | 0.9721 | 2.27 | 0.085 |
| **PSN-Partial** | | 0.3153 | 0.4728 | 0.9669 | 2.82 | 0.089 |
| Distance | $d_{\text{th}} = 1.5\text{m}$ | **0.3150** | 0.4824 | **0.9941** | 3.24 | 0.010 |
| **PSN-Full** | | **0.3108** | **0.4532** | 0.9717 | | |
| **PSN-Partial** | | 0.3152 | **0.4697** | 0.9665 | | |
| kNNs | | 0.3180 | 0.4768 | 0.9935 | | |
| Cost Evolution | $\lvert U^i \rvert = 2$ | 0.3378 | 0.4937 | 0.9849 | 3 | 0.097 |
| Gradient | | 0.3179 | 0.4733 | 0.9930 | | |
| Hessian | | 0.3191 | 0.4796 | 0.9926 | | |
| BF | | 0.3254 | 0.4794 | **0.9936** | | |
| CBF | | 0.3268 | 0.4885 | 0.9931 | | |
| CITR (Yang et al., 2019) | Parameter | ($\pm 0.0147$) | ($\pm 0.0257$) | ($\pm 0.0035$) | ($\pm 0.06$) | ($\pm 0.010$) |
| **PSN-Full** | $m_{\text{th}} = 0.5$ | 0.4987 | 0.4398 | **1.0000** | 1.34 | 0.049 |
| **PSN-Partial** | | 0.4940 | **0.4309** | **1.0000** | 1.50 | 0.050 |
| Distance | $d_{\text{th}} = 1.5\text{m}$ | 0.4986 | **0.4285** | 0.9851 | 2.13 | 0.080 |
| **PSN-Full** | | 0.4966 | 0.4398 | **1.0000** | | |
| **PSN-Partial** | | **0.4931** | 0.4438 | **1.0000** | | |
| kNNs | | 0.4975 | 0.4356 | 0.9765 | | |
| Cost Evolution | $\lvert U^i \rvert = 2$ | **0.4932** | 0.4413 | 0.8585 | 3 | 0.101 |
| Gradient | | 0.4987 | 0.4364 | 0.9785 | | |
| Hessian | | 0.4985 | 0.4367 | 0.9786 | | |
| BF | | 0.4986 | 0.4382 | 0.9745 | | |
| CBF | | 0.4952 | 0.4405 | 0.9723 | | |

* indicates the largest standard error of all methods for the corresponding metric.

### 4.1.2 Tasks and Metrics

We evaluate PSN on prediction and planning tasks. For prediction, we report *Average Displacement Error* (ADE ↓), *Final Displacement Error* (FDE ↓), and *Selection Consistency* (Consistency ↑), where consistency measures the temporal stability of the selected-player set rather than trajectory error. For planning, we report *Navigation Cost* (Nav. Cost ↓), *Collision Cost* (Col. Cost ↓), *Control Cost* (Ctrl. Cost ↓), *Selection Consistency* (Consistency ↑), *Minimum Distance* ($\text{Dist}_{\text{m}}$ ↑), *Trajectory Smoothness* ($\text{traj}_{\text{S}}$ ↓), and *Trajectory Length* ($\text{traj}_{\text{L}}$ ↓). We use ↑ for larger-is-better and ↓ for smaller-is-better. Metric implementations are provided in the Supplementary Material.

### 4.1.3 Baselines

We compare PSN Game against `Distance`, `kNNs`, `Gradient`, `Hessian`, `Cost Evolution`, `BF`, and `CBF` (Chahine et al., 2023). For PSN, we report two orthogonal design choices: input-information variants (`PSN-Full` and `PSN-Partial`) and selection-mode variants (`PSN-Threshold` and `PSN-Rank`). For brevity, full baseline definitions are provided in Appendix Section A.2.

### 4.1.4 Scenarios

To investigate the above hypotheses, we train both `PSN-Prediction` and `PSN-Planning` in an idealized setting for 4 and 10 agents with known ground-truth goals. At test time, PSN variants are benchmarked against baselines in a Monte Carlo study over 4 and 10 agent scenarios in prediction and planning tasks with ground truth goals (**Hypotheses 1 and 2**) and inferred goals (**Hypothesis 3**). We also evaluate the PSN that was trained for 10 agents in 20 agent scenarios to validate its scalability to a larger number of agents (**Hypothesis 4**) and finally, to real pedestrian trajectories in the CITR (Yang et al., 2019) dataset to assess generalization beyond the training distribution (where human motions are not governed by an explicit test-time game model) (**Hypothesis 5**). Scenario-specific details are in the Appendix.

## 4.2 Qualitative Analysis

*Advantages of PSN Game*: Table 1 summarizes the required information for runtime operation and parameters for different player selection methods, including agents' latest position (Pos.), latest velocity (Vel.), latest control (Ctrl.), trajectory history (Traj.), and game cost/constraint (Game). Our method offers several advantages: i) PSN-Partial requires only position information, which is more accessible compared to velocities and control inputs; ii) PSN Game reasons about agents' importance based on trajectory histories, not just instantaneous states, promoting better long-term consistency; iii) while PSN Game does require a model of agents' cost and constraints during training, that information is not used at runtime; iv) PSN Game requires no environment-specific parameter tuning at runtime. These properties enhance the versatility and general applicability of our framework.

*Advantages of Threshold-Based Methods over Ranking-Based Methods*: As a threshold-based method, PSN Game provides several advantages over ranking-based methods. Choosing an appropriate threshold (e.g., $m_{\text{th}} = 0.5$) is often easier to transfer across scenarios than fixing the number of selected players (See Appendix D for a detailed explanation). Thresholding also allows the number of selected players to adapt naturally to environmental density, unlike ranking-based methods that enforce a fixed number. Trajectory visualization in Fig. 3 illustrates that ranking-based methods may force the ego agent to select irrelevant players to meet a fixed quota, leading to unnecessary computation. Finally, threshold-based methods can be converted to ranking-style selections if needed, but not vice versa due to metric variability across environments.

## 4.3 Quantitative Result Analysis

### 4.3.1 Runtime Analysis

We conduct our experiment on a Desktop with an AMD Ryzen 9 7950X 16-Core Processor (128 GiB RAM) and an NVIDIA GeForce RTX 4090 GPU. In 4-player scenarios, the mean runtime is $3.15 \times 10^{-4}$s for GIN and $4.55 \times 10^{-4}$s for PSN, and in 10-player scenarios, the mean runtime is $6.72 \times 10^{-4}$s for GIN and $9.09 \times 10^{-4}$s for PSN. Such fast inference supports the real-time player selection in game-theoretic planning.

### 4.3.2 Prediction Test

Table 2 lists PSN-Prediction's performance compared with baseline methods in complete-information games with 4, 10, and 20 agents, and on real-world pedestrian trajectory data (CITR). Across these four settings and two prediction metrics (ADE and FDE), PSN remains competitive and is often among the stronger-performing methods. Performance differences across methods are often modest, which is expected because many methods select similar numbers of agents.

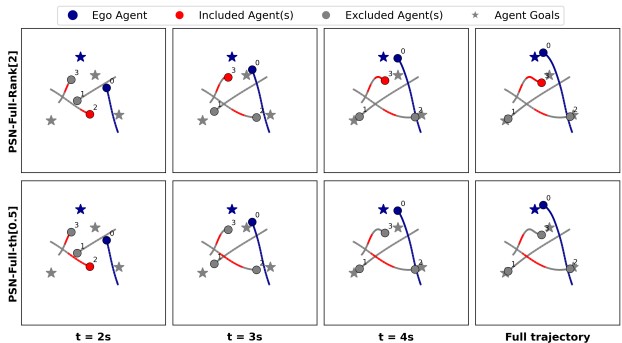

Figure 3: Trajectory visualization for PSN-Full-Rank (Top), PSN-Full-Threshold (Middle), and All (Bottom) in a 4 agent scenario. The ego agent (blue) iteratively solves the masked game that includes selected players (red) from the PSN and excludes irrelevant players (gray) to obtain an equilibrium strategy.

### 4.3.3 Planning Test

Table 3 lists the performance of PSN-Planning compared with baseline methods in a Monte Carlo study of complete-information games with 4 and 10 players. Trained via minimizing game cost for the ego-agent, PSN remains competitive across planning metrics and is often among the stronger-performing methods. Notably, PSN achieves the best mean values in both safety metrics (Collision Cost and Minimum Distance) in both scenarios, with only modest safety degradation relative to the full-player game, suggesting that PSN identifies safety-relevant agents reliably. The high selection-consistency values across scenarios also suggest that thresholding the soft mask did not produce rapid temporal flicker in our experiments.

Table 3: Bootstrapped mean planning performance in 4 and 10 agent scenarios with ground truth goals

| 4 Agents | Parameter | traj$_S$ ↓ ($\pm0.0019$) | traj$_L$ [m] ↓ ($\pm0.0741$) | Consistency ↑ ($\pm0.0035$) | Dist$_m$ [m] ↑ ($\pm0.0558$) | Num.P ($\pm0.04$) | Solving time [s] ($\pm0.008$) |
|---|---|---|---|---|---|---|---|
| **PSN-Full** | $m_{th} = 0.5$ | **0.0088** | 1.8934 | **0.9942** | 0.7969 | 1.32 | 0.045 |
| **PSN-Partial** | | 0.0097 | 1.9075 | 0.9927 | 0.7939 | 1.32 | 0.045 |
| Distance | $d_{th} = 1.0\,\text{m}$ | **0.0082** | 1.9006 | **0.9949** | 0.7860 | 1.20 | 0.040 |
| **PSN-Full** | | 0.0110 | 1.9168 | 0.9941 | **0.8173** | | |
| **PSN-Partial** | | 0.0150 | 1.9293 | 0.9926 | **0.8105** | | |
| kNNs | | 0.0101 | 1.9024 | 0.9888 | 0.7875 | | |
| Cost Evolution | $|U^i| = 1$ | 0.0125 | 1.9255 | 0.9800 | 0.7862 | 2 | 0.077 |
| Gradient | | 0.0098 | 1.8993 | 0.9885 | 0.7881 | | |
| Hessian | | 0.0095 | 1.9027 | 0.9914 | 0.7873 | | |
| BF | | 0.0098 | **1.8980** | 0.9918 | 0.7881 | | |
| CBF | | 0.0095 | 1.9027 | 0.9930 | 0.7863 | | |
| All | — | 0.0152 | 1.9437 | 1 | 0.8501 | 4 | 0.142 |

| 10 Agents | Parameter | traj$_S$ ↓ ($\pm0.0031$) | traj$_L$ [m] ↓ ($\pm0.1316$) | Consistency ↑ ($\pm0.0013$) | Dist$_m$ [m] ↑ ($\pm0.0423$) | Num.P ($\pm0.11$) | Solving time [s] ($\pm0.015$) |
|---|---|---|---|---|---|---|---|
| **PSN-Full** | $m_{th} = 0.5$ | 0.0189 | 2.4682 | 0.9920 | **0.5043** | 2.75 | 0.095 |
| **PSN-Partial** | | 0.0221 | 2.4924 | 0.9885 | 0.5035 | 2.88 | 0.098 |
| Distance | $d_{th} = 1.5\,\text{m}$ | **0.0153** | 2.4658 | **0.9931** | 0.4770 | 2.29 | 0.082 |
| **PSN-Full** | | 0.0213 | 2.4738 | 0.9919 | **0.5155** | | |
| **PSN-Partial** | | 0.0231 | 2.4994 | 0.9884 | 0.4982 | | |
| kNNs | | 0.0169 | **2.4514** | 0.9908 | 0.4801 | | |
| Cost Evolution | $|U^i| = 2$ | 0.0169 | 2.4931 | 0.9775 | 0.4848 | 3 | 0.101 |
| Gradient | | 0.0178 | 2.4486 | 0.9930 | 0.4816 | | |
| Hessian | | 0.0177 | 2.4560 | 0.9928 | 0.4820 | | |
| BF | | 0.0160 | 2.4560 | 0.9923 | 0.4792 | | |
| CBF | | **0.0155** | 2.4559 | **0.9932** | 0.4798 | | |
| All | — | 0.0215 | 2.4900 | 1 | 0.5680 | 10 | 0.353 |

### 4.3.4 Adaptation to a large number of agents

We also validate PSN's ability to adapt to a larger number of agents than encountered during training. The 20-agent prediction results are included in Table 2. Because PSN has a fixed input size, this experiment first applies a simple pre-filter to construct a candidate set of size 10. In this study, we use nearest-neighbor pre-filtering for simplicity; this should be viewed as an input-regularization step rather than as a claim that

kNN alone is sufficient for final player selection. We then apply PSN within this candidate set. Results remain competitive in this larger setting, suggesting robust adaptation when the number of agents exceeds training-time conditions. These results suggest that a PSN trained for 10-agent settings can remain effective in a larger 20-agent setting when paired with a simple candidate-set pre-filter.

**Remark 2.** We claim that any heuristic selection methods from the baselines Chahine et al. (2023) can be used as pre-filtering, while the exact choice of filters does not affect the selection quality, as it is only used to regulate the input size to PSN. The competitive performance of the combination of pre-filtering and PSN supports Hypothesis 4 that PSN readily apply to scenarios with more agents without retraining or fine-tuning, while retraining for large number of agents is often intractable.

### 4.3.5  Adaptation to pedestrian trajectory data

To evaluate PSN's adaptation to a non-game-theoretic scenario, we apply PSN Game to the CITR dataset (Yang et al., 2019), which involves 10 agents. The CITR rows in Table 2 show that PSN achieves the best ADE, while the best FDE is achieved by Distance; PSN remains competitive on both metrics. Notably, PSN still provides better ADE than solving a full-player game, suggesting that the learned selection rule transfers beyond the training dynamics. This outcome suggests that restricting attention to a small set of nearby or interaction-relevant neighbors can be sufficient for accurate prediction on this dataset.

### 4.3.6  Adaptation to incomplete-information games

Table 4: Bootstrapped mean prediction performance in 4 and 10 agent scenarios with **inferred goals**

| **4 Agents** | Parameter | ADE [m] ↓ ($\pm$0.0147) | FDE [m] ↓ ($\pm$0.0257) | Consistency ↑ ($\pm$0.0035) | Num.P ($\pm$0.06) | Solving time [s] ($\pm$0.010) |
|---|---|---|---|---|---|---|
| **PSN-Full** | $m_{\text{th}} = 0.5$ | **0.2989** | **0.5075** | 0.9903 | 1.66 | 0.061 |
| **PSN-Partial** | | 0.3150 | 0.5156 | 0.9870 | 1.88 | 0.070 |
| Distance | $d_{\text{th}} = 1\text{m}$ | 0.3200 | 0.5231 | **0.9941** | 2 | 0.080 |
| **PSN-Full** | | **0.3003** | 0.5037 | 0.9904 | | |
| **PSN-Partial** | | 0.3127 | 0.5139 | 0.9869 | | |
| kNNs | | 0.3118 | 0.5211 | 0.9899 | | |
| Cost Evolution | $|U^i| = 1$ | 0.3096 | 0.5081 | 0.9806 | 2 | 0.078 |
| Gradient | | 0.3121 | 0.5177 | 0.9791 | | |
| Hessian | | 0.3157 | 0.5211 | 0.9912 | | |
| BF | | 0.3127 | 0.5223 | 0.9898 | | |
| CBF | | 0.3149 | 0.5234 | **0.9922** | | |
| **10 Agents** | Parameter | ($\pm$0.0147) | ($\pm$0.0257) | ($\pm$0.0035) | ($\pm$0.06) | ($\pm$0.011) |
| **PSN-Full** | $m_{\text{th}} = 0.5$ | 0.4772 | 0.8094 | **0.9936** | 2.09 | 0.081 |
| **PSN-Partial** | | 0.4794 | 0.8214 | 0.9909 | 2.65 | 0.093 |
| Distance | $d_{\text{th}} = 1.5\text{m}$ | 0.4780 | 0.8083 | **0.9935** | 2.32 | 0.084 |
| **PSN-Full** | | **0.4697** | **0.8054** | 0.9935 | | |
| PSN-Partial | | 0.4782 | 0.8101 | 0.9910 | | |
| kNNs | | 0.4830 | 0.8272 | 0.9909 | | |
| Cost Evolution | $|U^i| = 2$ | 0.4679 | 0.8002 | 0.9783 | 3 | 0.105 |
| Gradient | | 0.4833 | 0.8323 | 0.9899 | | |
| Hessian | | 0.4839 | 0.8289 | 0.9919 | | |
| BF | | 0.4835 | 0.8274 | 0.9921 | | |
| CBF | | 0.4843 | 0.8343 | 0.9930 | | |

To validate PSN's adaptation to incomplete-information games where the agents' intentions are unknown, we test all methods aided with the trained goal inference network in 4 and 10-agent scenarios. Tables 4 and 5 report prediction and planning results for both 4-agent and 10-agent scenarios with inferred goals. PSN ranks among the top two methods on both prediction metrics and among the top three on all planning metrics, including first place in both safety metrics. This suggests that inferred goals are sufficiently accurate to support competitive PSN performance in incomplete-information scenarios.

## 5  Conclusion

In this work, we propose PSN Game, a game-theoretic planning framework that uses a Player Selection Network to improve scalability in multi-agent prediction and planning. Experiments in simulated environments and real-world pedestrian trajectory datasets show that PSN Game is competitive with, and often

Table 5: Bootstrapped mean planning performance in 4 and 10 agent scenarios with **inferred goals**

| 4 Agents | Parameter | traj$_S$ ↓ (±0.0023) | traj$_L$ [m] ↓ (±0.0805) | Consistency ↑ (±0.0032) | Dist$_m$ [m] ↑ (±0.0545) | Num.P (±0.04) | Solving time [s] (±0.010) |
|---|---|---|---|---|---|---|---|
| **PSN-Full** | $m_{\text{th}} = 0.5$ | **0.0087** | 1.8883 | **0.9942** | 0.7968 | 1.32 | 0.045 |
| **PSN-Partial** | | 0.0099 | 1.8898 | 0.9925 | 0.7980 | 1.32 | 0.045 |
| Distance | $d_{\text{th}} = 1.0\,\text{m}$ | 0.0086 | 1.8923 | 0.9950 | 0.7792 | 1.20 | 0.040 |
| **PSN-Full** | | 0.0110 | 1.9184 | 0.9940 | **0.8127** | | |
| **PSN-Partial** | | 0.0096 | 1.9003 | 0.9920 | 2 | | |
| kNNs | | 0.0117 | **1.7555** | 0.9898 | 0.8012 | | |
| Cost Evolution | | 0.0124 | 1.9196 | 0.9804 | 0.7829 | | |
| Gradient | $|U^i| = 1$ | 0.0098 | 1.8922 | 0.9885 | 0.7842 | 2 | 0.077 |
| Hessian | | 0.0113 | 1.7533 | 0.9912 | **0.8032** | | |
| BF | | 0.0095 | 1.8928 | 0.9915 | 0.7875 | | |
| CBF | | 0.0094 | 1.8963 | 0.9931 | 0.7846 | | |
| All | — | 0.0162 | 1.9352 | 1 | 0.8495 | 4 | 0.142 |
| **10 Agents** | Parameter | (±0.0029) | (±0.1301) | (±0.0015) | (±0.0485) | (±0.11) | (±0.015) |
| **PSN-Full** | $m_{\text{th}} = 0.5$ | 0.0196 | 2.4852 | 0.9920 | **0.4928** | 2.74 | 0.092 |
| **PSN-Partial** | | 0.0188 | 2.4579 | 0.9919 | 0.4865 | 2.74 | 0.092 |
| Distance | $d_{\text{th}} = 1.5\,\text{m}$ | **0.0169** | 2.4548 | 0.9930 | 0.4772 | 2.29 | 0.082 |
| **PSN-Full** | | 0.0218 | 2.5021 | 0.9919 | 0.5081 | | |
| **PSN-Partial** | | 0.0192 | 2.4601 | 0.9919 | 0.4853 | | |
| kNNs | | 0.0180 | 2.4503 | 0.9910 | 0.4796 | | |
| Cost Evolution | | 0.0183 | 2.4960 | 0.9767 | 0.4872 | | |
| Gradient | $|U^i| = 2$ | 0.0178 | 2.4478 | 0.9908 | 0.4797 | 3 | 0.101 |
| Hessian | | 0.0177 | 2.4560 | **0.9926** | 0.4820 | | |
| BF | | 0.0173 | **2.4492** | 0.9918 | 0.4794 | | |
| CBF | | **0.0165** | 2.4559 | **0.9932** | 0.4828 | | |
| All | — | 0.0236 | 2.5083 | 1 | 0.5665 | 10 | 0.353 |

outperforms, explicit player-selection methods while solving much smaller games than full-player approaches. Its flexible data requirements and compatibility with existing planning pipelines make it a scalable, efficient, and safety-aware solution. We also identify future directions, including richer nonlinear dynamics, stronger learning-based baselines, and temporal smoothing for deployment robustness.

**Acknowledgments**

This work was supported by the National Science Foundation under grants 2336840 and 2409535, and by the Army Research Laboratory under cooperative agreements W911NF-23-2-0011 and W911NF-25-2-0021.

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

# A  Training Details

## A.1  Network Architecture

We describe the architectures of GIN and PSN as follows:

**Goal Inference Network**: Given the observation of all agents' trajectories $\mathbf{x}_{0:K} \in \mathbb{R}^{(K+1) \times N \times d}$ ($d = 2$ for partial and $d = 4$ for full), we encode each agent's $T$-step sequence with a GRU (hidden size $H = 64$), keep the final hidden states $\{\mathbf{z}_i\}_{i=1}^N$, and concatenate them into $\mathbf{z} \in \mathbb{R}^{NH}$. An MLP with hidden sizes $256 \to 128 \to 32$ (ReLU; dropout $p = 0.3$ between hidden layers) maps $\mathbf{z}$ to a linear output of size $2N$, reshaped to $\hat{p}_g \in \mathbb{R}^{N \times 2}$ giving $\hat{p}_g^i$ for each agent.

**Player Selection Network**: Given the observation of all agents' trajectories $\mathbf{x}_{0:K} \in \mathbb{R}^{(K+1) \times N \times d}$, the PSN predicts an ego-centric, continuous selection mask $m^i \in [0,1]^{N-1}$. As in the goal model, per-agent GRUs (hidden size 64) encode sequences; their final states are concatenated and passed through an MLP $256 \to 128 \to 32$ (ReLU; dropout 0.3), followed by a sigmoid layer to produce $m^i$. A pretrained goal-inference network supplies per-agent goals for a differentiable iLQR game.

## A.2  Baseline Definitions

In experiments, an agent is selected by each method according to the following rule:

i. `PSN-Threshold` (ours): the agent's mask $m^{ij}$ is larger than a predefined threshold $m_{\text{th}}$.

ii. `PSN-Rank` (ours): the agent has one of the top $|U^i|$ largest mask values $m^{ij}$.

iii. `Distance`: the agent's current distance from the ego agent is less than a predefined radius $r_{\text{th}}$.

iv. `kNNs`: the agent is among the top $|U^i|$ closest agents to the ego agent in Euclidean distance.

v. `Gradient` (Chahine et al., 2023): the agent has one of the top $|U^i|$ largest norms of the gradient of the ego agent's shared cost with respect to that agent's current control input.

vi. `Hessian` (Chahine et al., 2023): the agent has one of the top $|U^i|$ largest norms of the Hessian of the ego agent's shared cost with respect to that agent's current control input.

vii. `Cost Evolution` (Chahine et al., 2023): the agent causes one of the top $|U^i|$ largest increases in the ego agent's shared cost over the last step.

viii. `BF` (Chahine et al., 2023): the agent has one of the top $|U^i|$ smallest barrier function values encoding the safety constraint.

ix. `CBF` (Chahine et al., 2023): the agent has one of the top $|U^i|$ smallest control barrier function values encoding the safety constraint.

## A.3  Scenario Details

We list the scenario details and scenario-specific game constraints/weights in Table 6. Note that the CITR dataset collects trajectories from 10 pedestrians, which is a non-game-theoretic setting.

Table 6: Scenario Details

|  | 4 Agent | 10 Agent | 20 Agent | CITR (Yang et al., 2019) |
|---|---|---|---|---|
| Scenario size | $5\,\text{m} \times 5\,\text{m}$ | $7\,\text{m} \times 7\,\text{m}$ | $7\,\text{m} \times 7\,\text{m}$ | $7.5\,\text{m} \times 25.5\,\text{m}$ |
| Observation interval |  | 10 Steps = 1 s |  |  |
| Receding horizon steps |  | 50 Steps = 5 s |  | Vary by data |
| Game weight equation 4.1.1 |  | $\mathbf{w}^i = [0.1, 0.001, 0.1, 0.1]$ |  |  |

### A.4 Training Data and Parameters

We train four variants of our GIN and PSN (Full, Partial) in the 4 agent scenario and 10 agent scenario mentioned above, respectively. Training parameters for each variant are listed in Table 7 and 8 with randomly generated initial states and goal positions, we use a differentiable game solver (Sun, 2025) to generate all agents' trajectories $\mathbf{x}$ from a full-player game where all the other agents are considered by the ego agent.

Table 7: Training parameters for GIN

| Scenario | 4 Agent | | 10 Agent | |
|---|---|---|---|---|
| Method | Full | Partial | Full | Partial |
| Input size | 160 | 80 | 400 | 200 |
| Learning rate | $1 \times 10^{-3}$ | | | |
| Batch size | 32 | | | |
| Epochs | 100 | | | |

Table 8: Training parameters for PSN

| Scenario | 4 Agent Scenario | | 10 Agent | |
|---|---|---|---|---|
| Method | Full | Partial | Full | Partial |
| Input size | 160 | 80 | 400 | 200 |
| Learning rate | $1 \times 10^{-3}$ | | | |
| Batch size | 32 | | | |
| Epochs | 100 | | | |
| Loss weight | $\sigma_1 = 0.075, \sigma_2 = 0.075$ for prediction $\sigma_1 = 0.5, \sigma_2 = 0.5$ for planning | | | |

## B Evaluation Metrics

We provide the mathematical formulation of evaluation metrics mentioned in Section 4.1 as follows, for the ego agent $i$:

$$\text{ADE} \downarrow := \sum_{t=K}^{K+T} \|p_t^i - p_{t,\text{gt}}^i\|, \quad \text{FDE} \downarrow := \|p_{K+T}^i - p_{K+T,\text{gt}}^i\|,$$

$$\text{Nav. Cost} \downarrow := \sum_{t=K}^{K+T} \|p_k^i - p_{\text{ref},k}^i\|_2^2, \quad \text{Col. Cost} \downarrow := \sum_{\substack{j=1 \\ j \neq i}}^{N} \exp\left(-\|p_k^i - p_k^j\|_2^2\right), \quad \text{Ctrl. Cost} \downarrow := \sum_k \|u_k^i\|_2^2,$$

$$\text{traj}_S \downarrow := \sum_k \left\| \frac{p_k^i - p_{k-1}^i}{\|p_k^i - p_{k-1}^i\|_2} - \frac{p_{k-1}^i - p_{k-2}^i}{\|p_{k-1}^i - p_{k-2}^i\|_2} \right\|_2, \quad \text{traj}_L \downarrow := \sum_k \|p_k^i - p_{k-1}^i\|_2,$$

$$\text{Dist}_m \uparrow := \min_k \|p_k^i - p_k^{\neg i}\|_2, \quad \text{Consistency} \uparrow := \sum_k \left(1 - \frac{\|M_k^i - M_{k-1}^i\|_1}{N-1}\right).$$

Lower ADE and FDE indicate more accurate predictions. Lower Nav. Cost, Col. Cost, and Ctrl. Cost indicate better planning performance. Lower $\text{traj}_S$ indicates a gentler cumulative change in direction, thus it is more desirable. Lower $\text{traj}_L$ indicates fewer detours for the ego agent to approach the goal. Higher $\text{Dist}_m$ indicates better safety performance. Higher Consistency indicates fewer changes in player selection, which also promotes a smoother trajectory and shorter trajectory length.

## C Full Planning Tables

For completeness, we provide the full versions of the planning tables from the main text (including Nav. Cost, Col. Cost, and Ctrl. Cost). To improve readability, both tables are rotated 90° counterclockwise.

Table 9: Full version of Table 3 with all planning metrics (including Nav. Cost, Col. Cost, and Ctrl. Cost).

| 4 Agents | Parameter | Nav. Cost ↓ (±0.1359) | Col. Cost ↓ (±0.0764) | Ctrl. Cost ↓ (±0.0045) | trajS ↓ (±0.0019) | trajL [m] ↓ (±0.0741) | Consistency ↑ (±0.0035) | Dist_m [m] ↑ (±0.0558) | Num.P (±0.04) | Solving time [s] (±0.008) |
|---|---|---|---|---|---|---|---|---|---|---|
| **PSN-Full** | $m_{\mathrm{th}} = 0.5$ | 3.2410 | 1.9387 | **0.1147** | **0.0088** | **1.8934** | **0.9942** | 0.7969 | 1.32 | 0.045 |
| **PSN-Partial** | | 3.2306 | 1.9410 | 0.1157 | 0.0097 | 1.9075 | 0.9927 | 0.7939 | 1.32 | 0.045 |
| Distance | $d_{\mathrm{th}} = 1.0\,\mathrm{m}$ | 3.2494 | 1.9517 | 0.1163 | **0.0082** | 1.9006 | **0.9949** | 0.7860 | 1.20 | 0.040 |
| **PSN-Full** | | 3.2336 | **1.9084** | 0.1175 | 0.0110 | 1.9168 | 0.9941 | **0.8173** | | |
| **PSN-Partial** | | **3.2284** | **1.8947** | 0.1181 | 0.0150 | 1.9293 | 0.9926 | **0.8105** | | |
| kNNs | | 3.2636 | 1.9267 | 0.1160 | 0.0101 | 1.9024 | 0.9888 | 0.7875 | | |
| Cost Evolution | $|U^i| = 1$ | **3.2167** | 1.9249 | 0.1189 | 0.0125 | 1.9255 | 0.9800 | 0.7862 | 2 | 0.077 |
| Gradient | | 3.2563 | 1.9281 | 0.1165 | 0.0098 | 1.8993 | 0.9885 | 0.7881 | | |
| Hessian | | 3.2536 | 1.9325 | 0.1164 | 0.0095 | 1.9027 | 0.9914 | 0.7873 | | |
| BF | | 3.2574 | 1.9287 | 0.1158 | 0.0098 | **1.8980** | 0.9918 | 0.7881 | | |
| CBF | | 3.2513 | 1.9304 | **0.1155** | 0.0095 | 1.9027 | 0.9930 | 0.7863 | | |
| All | — | 3.2962 | 1.7890 | 0.1148 | 0.0152 | 1.9437 | 1 | 0.8501 | 4 | 0.142 |

| 10 Agents | Parameter | Nav. Cost ↓ (±0.2173) | Col. Cost ↓ (±0.1535) | Ctrl. Cost ↓ (±0.0073) | trajS ↓ (±0.0031) | trajL [m] ↓ (±0.1316) | Consistency ↑ (±0.0013) | Dist_m [m] ↑ (±0.0423) | Num.P (±0.11) | Solving time [s] (±0.015) |
|---|---|---|---|---|---|---|---|---|---|---|
| **PSN-Full** | $m_{\mathrm{th}} = 0.5$ | 4.4737 | 3.8028 | 0.1511 | 0.0189 | 2.4682 | 0.9920 | **0.5043** | 2.75 | 0.095 |
| **PSN-Partial** | | 4.4273 | **3.7905** | 0.1531 | 0.0221 | 2.4924 | 0.9885 | 0.5035 | 2.88 | 0.098 |
| Distance | $d_{\mathrm{th}} = 1.5\,\mathrm{m}$ | **4.4041** | 3.8669 | 0.1488 | **0.0153** | 2.4658 | **0.9931** | 0.4770 | 2.29 | 0.082 |
| **PSN-Full** | | 4.4820 | 3.7913 | 0.1537 | 0.0213 | 2.4738 | 0.9919 | **0.5155** | | |
| **PSN-Partial** | | 4.4273 | **3.7875** | 0.1545 | 0.0231 | 2.4994 | 0.9884 | 0.4982 | | |
| kNNs | | 4.4517 | 3.8434 | 0.1496 | 0.0169 | **2.4514** | 0.9908 | 0.4801 | | |
| Cost Evolution | $|U^i| = 2$ | **4.3470** | 3.8311 | 0.1540 | 0.0169 | 2.4931 | 0.9775 | 0.4848 | 3 | 0.101 |
| Gradient | | 4.4392 | 3.8585 | 0.1487 | 0.0178 | 2.4560 | 0.9930 | 0.4816 | | |
| Hessian | | 4.4317 | 3.8416 | **0.1479** | 0.0177 | **2.4486** | 0.9928 | 0.4820 | | |
| BF | | 4.4501 | 3.8415 | 0.1484 | 0.0160 | 2.4560 | 0.9923 | 0.4792 | | |
| CBF | | 4.4294 | 3.8471 | **0.1479** | **0.0155** | 2.4559 | **0.9932** | 0.4798 | | |
| All | — | 4.7586 | 3.4682 | 0.1425 | 0.0215 | 2.4900 | 1 | 0.5680 | 10 | 0.353 |

Table 10: Full version of Table 5 with all planning metrics (including Nav. Cost, Col. Cost, and Ctrl. Cost).

| 4 Agents | Parameter | Nav. Cost ↓ (±0.1603) | Col. Cost ↓ (±0.0758) | Ctrl. Cost ↓ (±0.0046) | trajS → (±0.0023) | trajL [m] ↓ (±0.0805) | Consistency ↑ (±0.0032) | Dist$_m$ [m] ↑ (±0.0545) | Num.P (±0.04) | Solving time [s] (±0.010) |
|---|---|---|---|---|---|---|---|---|---|---|
| **PSN-Full** | $m_{th} = 0.5$ | 3.2451 | 1.9421 | **0.1145** | **0.0087** | 1.8883 | **0.9942** | 0.7968 | 1.32 | 0.045 |
| **PSN-Partial** | | 3.2483 | 1.9355 | 0.1151 | 0.0099 | 1.8898 | 0.9925 | 0.7980 | 1.32 | 0.045 |
| Distance | $d_{th} = 1.0\,m$ | 3.2477 | 1.9635 | 0.1160 | **0.0086** | 1.8923 | **0.9950** | 0.7792 | 1.20 | 0.040 |
| **PSN-Full** | | **3.2422** | **1.9103** | 0.1174 | 0.0110 | 1.9184 | 0.9940 | **0.8127** | | |
| **PSN-Partial** | | 3.2465 | 1.9403 | 0.1153 | 0.0096 | 1.9003 | 0.9920 | 0.8012 | | |
| kNNs | | 3.9276 | 1.9751 | **0.1145** | 0.0117 | **1.7555** | 0.9898 | 0.7829 | | |
| Cost Evolution | $|U^i| = 1$ | **3.2197** | 1.9309 | 0.1189 | 0.0124 | 1.9196 | 0.9804 | 0.7842 | 2 | 0.077 |
| Gradient | | 3.2684 | 1.9363 | 0.1161 | 0.0098 | 1.8922 | 0.9885 | 0.8032 | | |
| Hessian | | 3.9282 | 1.9831 | **0.1145** | 0.0113 | **1.7533** | 0.9912 | **0.8032** | | |
| BF | | 3.2669 | **1.9308** | 0.1155 | 0.0095 | 1.8928 | 0.9915 | 0.7875 | | |
| CBF | | 3.2606 | 1.9324 | 0.1151 | 0.0094 | 1.8963 | 0.9931 | 0.7846 | | |
| All | — | 3.3084 | 1.7948 | 0.1147 | 0.0162 | 1.9352 | 1 | 0.8495 | 4 | 0.142 |

| 10 Agents | Parameter | Nav. Cost ↓ (±0.2254) | Col. Cost ↓ (±0.1563) | Ctrl. Cost ↓ (±0.0066) | trajS → (±0.0029) | trajL [m] ↓ (±0.1301) | Consistency ↑ (±0.0015) | Dist$_m$ [m] ↑ (±0.0485) | Num.P (±0.11) | Solving time [s] (±0.015) |
|---|---|---|---|---|---|---|---|---|---|---|
| **PSN-Full** | $m_{th} = 0.5$ | 4.4480 | 3.8417 | 0.1530 | 0.0196 | 2.4852 | 0.9920 | **0.4928** | 2.74 | 0.092 |
| **PSN-Partial** | | 4.4301 | 3.8362 | 0.1499 | 0.0188 | 2.4579 | 0.9919 | 0.4865 | 2.74 | 0.092 |
| Distance | $d_{th} = 1.5\,m$ | **4.4085** | 3.8717 | 0.1491 | **0.0169** | 2.4548 | 0.9930 | 0.4772 | 2.29 | 0.082 |
| **PSN-Full** | | 4.4656 | **3.8169** | 0.1564 | 0.0218 | 2.5021 | 0.9919 | **0.5081** | | |
| **PSN-Partial** | | 4.4299 | 3.8399 | 0.1521 | 0.0192 | 2.4601 | 0.9919 | 0.4853 | | |
| kNNs | | 4.4557 | 3.8480 | 0.1500 | 0.0180 | 2.4503 | 0.9910 | 0.4796 | | |
| Cost Evolution | $|U^i| = 2$ | **4.3612** | **3.8284** | 0.1560 | 0.0183 | 2.4960 | 0.9767 | 0.4872 | 3 | 0.101 |
| Gradient | | 4.4537 | 3.8491 | 0.1487 | 0.0178 | **2.4478** | 0.9908 | 0.4797 | | |
| Hessian | | 4.4296 | 3.8858 | **0.1483** | 0.0177 | 2.4560 | **0.9926** | 0.4820 | | |
| BF | | 4.4407 | 3.8533 | 0.1485 | 0.0173 | **2.4492** | 0.9918 | 0.4794 | | |
| CBF | | 4.4379 | 3.8583 | **0.1482** | **0.0165** | 2.4559 | **0.9932** | 0.4828 | | |
| All | — | 4.7659 | 3.5107 | 0.1467 | 0.0236 | 2.5083 | 1 | 0.5665 | 10 | 0.353 |

# D   Mask Distribution Histograms

We report the distribution of player masks outputted by PSN-Prediction and PSN-Planning for 4, 10 and 20-player scenarios. These distributions clearly support our claim that the choice of selection threshold $m_{\text{th}}$ can be trivially set to 0.5, as all distributions have peaks around 0 and 1.

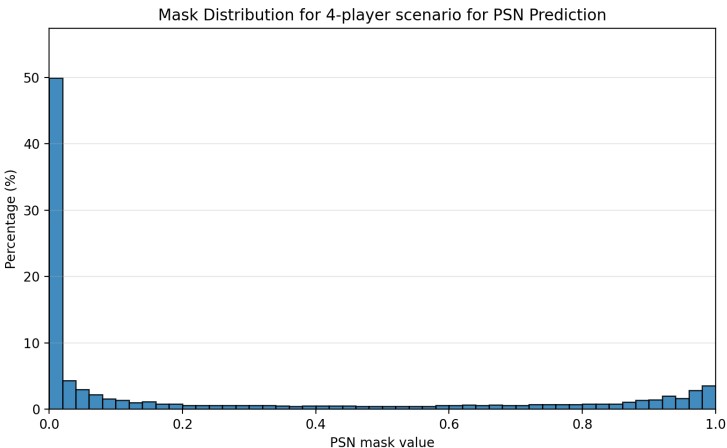

Figure 4: Mask Distribution for PSN-Prediction for 4-player scenarios

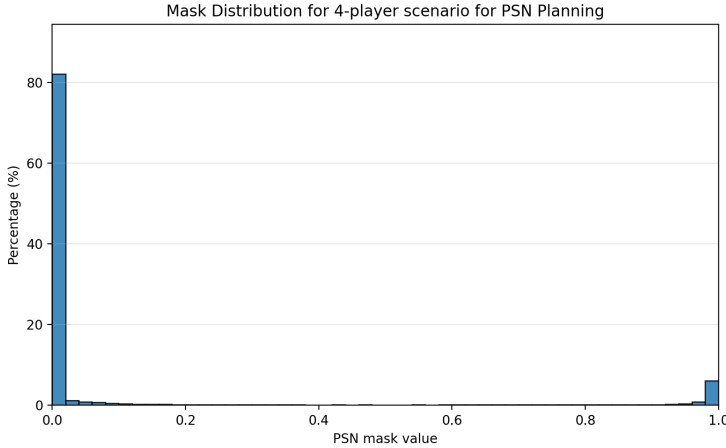

Figure 5: Mask Distribution for PSN-Planning for 4-player scenarios

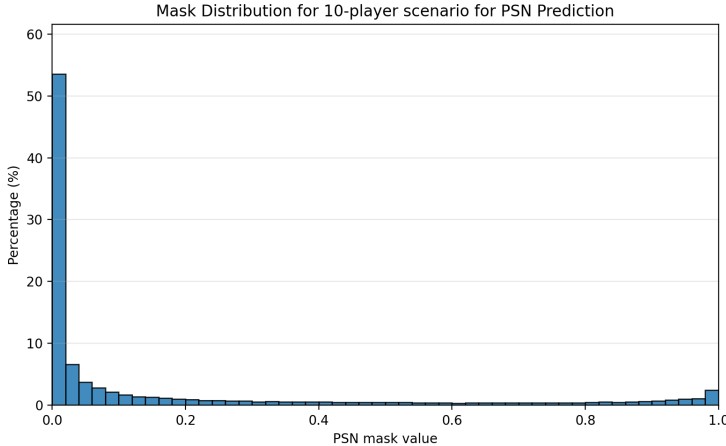

Figure 6: Mask Distribution for PSN-Prediction for 10-player scenarios

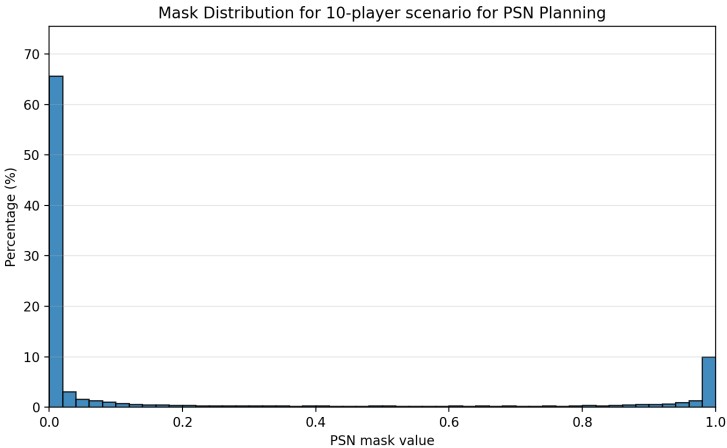

Figure 7: Mask Distribution for PSN-Planning for 10-player scenarios

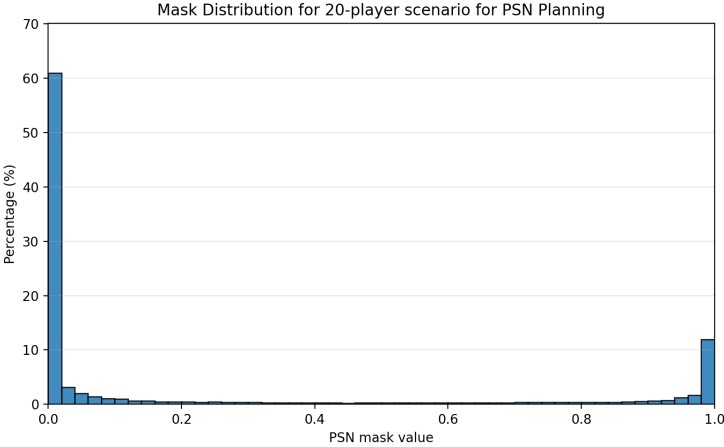

Figure 8: Mask Distribution for PSN-Prediction for 20-player scenarios

