# OpenReview forum: "A Player Selection Network for Scalable Game-Theoretic Prediction and Planning"
_TMLR — Accepted by TMLR_

### Review · Reviewer_NtnS · 2026-04-13

**Summary Of Contributions:**

The paper introduces PSN Game, a framework designed to scale game-theoretic planning for multi-agent systems. The core innovation is a Player Selection Network (PSN) that identifies influential agents to create a smaller, "masked" game, significantly reducing the $O(N^3)$ computational complexity of Nash equilibrium solvers.

Strengths:

Efficiency: Reduces game size by 50–75% while maintaining performance.
Flexibility: Does not require online parameter tuning or access to other agents' internal control/cost parameters at runtime.
Generalization: Demonstrates zero-shot scalability to 20 agents and transferability to real-world pedestrian datasets (CITR).

Weaknesses:
Dependency on Pre-filtering: Scalability to very large agent counts still relies on a simple kNN pre-filter to fit the PSN's fixed input size.
Dynamics Simplification: Evaluation is primarily focused on double-integrator dynamics, and more complex nonlinear behaviors are mentioned but not extensively tested.

**Additional Comments:**

The paper is well-structured, and the use of a differentiable solver to allow backpropagation through the selection mask is a technically sound and elegant approach to the selection problem. Therefore, I lean to accept this paper.

**Audience:**

Yes

**Audience Explanation:**

The findings are highly relevant to researchers in robotics, multi-agent systems, and autonomous driving. The method addresses a fundamental bottleneck, the cubic scaling of game-theoretic solvers, by applying a learning-based selection mechanism, which is a significant topic of interest for the machine learning and planning communities.

**Broader Impact Concerns:**

The paper focuses on safety and efficiency in multi-agent navigation, which generally aligns with the positive development of autonomous systems. No significant ethical risks were identified.

**Claims And Evidence:**

Yes

**Claims Explanation:**

The authors provide convincing evidence through a series of Monte Carlo simulations and real-world data tests. The PSN-Threshold variants consistently outperform or match traditional heuristics (like KNN or Distance-based selection) in prediction accuracy (ADE/FDE) and planning safety (Minimum Distance)

**Requested Changes:**

Critical: Clarify the sensitivity of the $m_{th}$ threshold. While $0.5$ is used, an analysis of how performance degrades if the threshold is significantly higher or lower would strengthen the claim of "no manual tuning".

Strengthening: Provide a brief discussion or small-scale test on how the framework handles nonlinear dynamics beyond the mention in the conclusion, as this is a key hurdle for real-world robotics.

Strengthening: Formally define the "simple pre-filter" used for the 20-agent case to ensure reproducibility of the scalability results.

Also see Weaknesses

---

> ### Author Response · Authors · 2026-05-22
> **Official Comment to Reviewer NtnS**
>
> Requested Changes:
> 1. We agree with the reviewer. The claim that PSN does not require online manual tuning should be supported by additional evidence. In Appendix D, we include mask distribution histograms of the learned mask values outputted by PSN in different scenarios. These plots show that the masks tend to concentrate near binary values, which would support the use of a fixed default threshold such as $m_\text{th}=0.5$.
>
> 2. We thank the reviewer for this suggestion. We agree that nonlinear dynamics are an important consideration for real-world robotic systems. We refer the reviewer to Page 8, where we have clarified that the current double-integrator experiments are intended as a controlled benchmark for studying the player-selection mechanism itself. The PSN framework is not inherently tied to double-integrator dynamics; it only requires that the downstream masked game can be solved by the chosen game solver.
>
> 3. We thank the reviewer for suggesting formally defining the simple pre-filter. The purpose of applying a pre-filter is to regulate the input trajectory size to fit the PSN. Therefore, if there are many more agents in the scenario than the number required by PSN, we use a pre-filter to eliminate less important agents coarsely, and then do a fine-filtering with PSN. We claim that any heuristic way of filtering (e.g., baseline methods) is suitable for pre-filtering, while in the manuscript, we use the simplest k nearest neighbor (kNN). Simulation results show that the combination of pre-filtering and PSN still outperforms the baseline methods in metrics. We add Remark 2 in the manuscript to explain and emphasize our claim.

---

### Review · Reviewer_mRVc · 2026-04-24

**Summary Of Contributions:**

The authors present a solution based on selecting a subset of players participating in a game and on a generalisation to games with incomplete information that masks players so that existing algorithms can work on a subset of the initial problem while keeping the same practical results. The developed methodology has the major benefit of choosing the number of selected players at runtime.

The paper shows that a rather simple idea can be very effective, which leads to a paper that spends most of its pages on evaluation (with ample details to allow for reproduction).

**Audience:**

Yes

**Audience Explanation:**

Application of learning to game theory, joining a large thread of papers on the same intersection of topics.

**Broader Impact Concerns:**

No need for an impact statement in my opinion: the paper is about a mathematical problem, with no foreseeable ethical issue.

**Claims And Evidence:**

Yes

**Claims Explanation:**

All the evidence is provided by the means of numerical experiments against many baselines.

**Requested Changes:**

Weaknesses (sorted by decreasing level of importance):
- The authors mention on page 6, section 3.5: "To keep evaluation focused, we apply PSN only to the ego agent when reporting prediction results in Section 4." However, such an evaluation would be very helpful to understand the impact of their methodology, as the article in its current form does not address the question of deploying the methodology to several or to all players in a given game (albeit such scenarios are considered in the introduction).
- Overall, the authors only use modest numbers of agents. In their introduction (section 1, page 1), they mention that a 5-player game is small, but only consider games up to 20 participants.
- In the evaluations, the authors only consider one value of m_{th}. It would be interesting to study its impact, even only on small-scale problems. Maybe there are trade-offs to be discovered by choosing a more appropriate value, as it is directly linked to the level of confidence of the PSN (due to the loss functions).

Minor remark:
- Table 2, page 9: the distinction between the two star symbols are hard to make (the authors could use \dag instead of one type of star).

---

> ### Author Response · Authors · 2026-05-22
> **Official Comment to Reviewer mRVc**
>
> W1: We thank the reviewer for this helpful suggestion. In the current evaluation, we apply PSN only to the ego agent in order to isolate the impact of the learned selector on the ego agent’s prediction or planning problem. If PSN is applied simultaneously to all players, the resulting rollout becomes a fully decentralized masked-game procedure, which is an important deployment setting but also introduces additional coupled effects that make it harder to isolate the contribution of the ego-agent selector.
>
> W2: To the best of our knowledge, most literature about game-theoretic planning and prediction considers a number of agents lower than 10 [1,2]. And other learning-based methods [3,4] also utilize a pooling mechanism or pre-filtering to regulate the number of considered agents to at most 10. Therefore, we think the scale of 20 agents is sufficiently large.
>
> W3: We thank the reviewer for the suggestion. In Appendix D, we include mask distribution histograms of the learned mask values output by PSN in different scenarios. These plots show that the masks tend to concentrate near binary values, which would support the use of a fixed default threshold such as $m_\text{th}=0.5$.
>
> W4: We thank the reviewer for the suggestion. We have modified the table contents so it is no longer a concern.
>
> References:
>
> [1] M. Chahine, R. Firoozi, W. Xiao, M. Schwager, and D. Rus, “Local non-cooperative games with principled
> player selection for scalable motion planning,” in 2023 IEEE/RSJ International Conference on Intelligent
> Robots and Systems (IROS). IEEE, 2023, pp. 880–887.
>
> [2] T. Qiu and D. Fridovich-Keil, “Inferring occluded agent behavior in dynamic games from noise corrupted
> observations,” IEEE Robotics and Automation Letters, 2024.
>
> [3] S. Shi, L. Jiang, D. Dai, and B. Schiele, “Motion transformer with global intention localization and local
> movement refinement,” Advances in Neural Information Processing Systems, vol. 35, pp. 6531–6543, 2022.
>
> [4] A. Alahi, K. Goel, V. Ramanathan, A. Robicquet, L. Fei-Fei, and S. Savarese, “Social lstm: Human
> trajectory prediction in crowded spaces,” in Proceedings of the IEEE conference on computer vision and
> pattern recognition, 2016, pp. 961–971.

---

### Review · Reviewer_7XZv · 2026-05-08

**Summary Of Contributions:**

**The contributions of the paper**:

The authors aim to address the scalability issue of game-theoretic planning in multi-agent environments. They propose combining the Player Selection Network (PSN) and the Goal Inference Network (GIN) to dynamically reduce the size of the game that the ego agent solves. Through this approach, they present a learning-based framework that reduces computational complexity while maintaining prediction accuracy and planning safety.

**Strengths:**
- From a presentation perspective, the authors attempted to present the framework structure through Figure 1 and Algorithms 1 and 2.
- From a soundness perspective, the end-to-end learning structure proposed by the authors enables backpropagation through the mask, and the approach in Remark 1 is also technically reasonable.
- From an originality perspective, whereas prior work has been limited to heuristic-based selection, an end-to-end framework that learns task-relevant masks from past trajectories can be considered original in this field.

**Weaknesses:**
- From a presentation perspective, there is a weakness in that the gap between the claims related to "scalability" and the actual experimental scope may mislead readers. While the Introduction and Abstract strongly emphasize the need for large-scale, real-time systems, the experiments are limited to at most 20 agents. Furthermore, the Related Work section provides insufficient coverage of the learning-based multi-agent prediction literature beyond player selection works.
- From a soundness perspective, it is difficult to discern a quantitative connection between the reduction in the number of players and actual computation time reduction.
- From a significance perspective, there are several weaknesses as follows.
  - From the perspective of the machine learning community at TMLR, the learning-theoretic contribution of the proposed PSN architecture itself is limited.
  - Additionally, the experimental analysis of computational efficiency is insufficient.
- From an originality perspective, the originality of the paper lies more in the problem formulation than in the architecture; however, the depth of originality is limited by the absence of theoretical analysis supporting it — for example, under what conditions does the PSN converge to the optimal mask. In particular, it remains unclear what the PSN actually learns and what new insights the authors' proposed method provides.

**Audience:**

Yes

**Audience Explanation:**

Researchers in this field would find the idea of learning-based player selection practically relevant. This is because reducing game complexity without sacrificing planning performance is a widely recognized bottleneck in deploying such frameworks at scale.

**Claims And Evidence:**

Yes

**Claims Explanation:**

Although I marked “yes” above, my opinion falls somewhere between “yes” and “no.”
In my view, the authors’ claim that their method reduces computation time by solving smaller-scale games is not directly supported by actual runtime measurements.
Instead, the authors report only the number of players, as a proxy metric, which does not constitute compelling evidence.
Nevertheless, the authors verify the method’s strong performance by examining prediction and planning metrics from various perspectives.

**Requested Changes:**

1. What is the computational overhead of the PSN forward pass itself, and how does it compare to the game solver runtime?
2. Figure 2 shows cubic scaling from Fridovich-Keil's solver, but this is not measured for the PSN Game itself. Can the authors provide a similar scaling plot for their full pipeline?
3. In Eq. (6), the loss weights $\sigma_{1:4}$ differ between prediction and planning settings. What is the sensitivity of performance to these values, and how were they chosen?
4. In the 20-agent experiment, why is kNN pre-filtering applied before PSN? How sensitive are the results to the choice of pre-filter?
5. In Algorithm 1 as well, is GIN called only once at initialization, or is it updated at every step within the receding horizon loop? The text specifies "once at initialization" only for Algorithm 2, so the frequency of GIN calls in Algorithm 1 is unclear.
6. In Line 3 of Algorithm 1, is $\hat{x}_{\{k-K:k\}}, used as the input to PSN-Full, an observed state?
7. What does the star (★) notation in Figure 3 represent?
8. Section 4.2 claims that PSN generates "smoother, shorter ego trajectories," but this is difficult to visually verify in Figure 3. Can the authors provide quantitative evidence?

---

> ### Author Response · Authors · 2026-05-22
> **Official Comment to Reviewer 7XZv**
>
> W1: To the best of our knowledge, most literature about game-theoretic planning and prediction considers a number of agents lower than 10 [1,2]. And other learning-based methods [3,4] also utilize a pooling mechanism or pre-filtering to regulate the number of considered agents to at most 10. Therefore, we think the scale of 20 agents is sufficiently large. We thank the reviewer for suggesting adding learning-based multi-agent prediction literature. However, we clarify that our prediction is still conducted based on a game-theoretic mechanism where a dynamic game is solved to output the predicted trajectories. We think such formulation is distinct from those learning-based frameworks, and that's the reason we did not include the comparison with those methods.
>
> W2: We have included the time statistics in all tables that directly reflect the increase in computational efficiency resulting from the decrease in agent numbers.
>
> W3: We agree that the main contribution of the paper is not a novel neural network architecture by itself. The originality of the work lies in formulating player selection as a downstream game-aware learning problem and using the learned selection mask to reduce the size of the game solved during prediction and planning. We emphasize that PSN is trained through a differentiable dynamic game solver, so the mask is learned according to the effect of selected players on the downstream game solution rather than from manually labeled important players. This distinguishes the method from standard neural-network-based prediction architectures. We also agree that the computational efficiency analysis should be strengthened. We have added actual runtime measurements in all tables and report the PSN runtime. Since PSN inference is very fast compared with solving the game, we expect the end-to-end runtime to be reduced when the selected subset is substantially smaller than the full player set.
>
> W4: We agree that the current paper does not provide a theoretical guarantee that PSN converges to the globally optimal mask. The mask-selection problem is combinatorial, and the neural-network training problem is nonconvex, so we do not claim global optimality. For a finite number of players, one could, in principle, define an optimal mask with respect to a chosen downstream loss by exhaustively evaluating all possible masks. However, this requires searching over exponentially many subsets and is therefore impractical for online receding-horizon prediction or planning. PSN is designed as an amortized approximation to this selection process: it learns from trajectory observations to produce masks that preserve the downstream game solution while encouraging sparsity. We clarify that PSN learns to identify agents whose presence materially affects the ego agent’s predicted trajectory or planning cost, while excluding agents whose inclusion primarily increases computation.
>
> Requested Changes:
> 1. We report the runtime for PSN and GIN in Section 4.3.1. Results show that the runtime of PSN and GIN is ignorable compared to the game-solving part, resulting from their light-weight GRU structures, and supports the real-time application.
> 2. The main purpose of Fig 2 is to demonstrate the $O(n^3)$ complexity for game-solving. We refer the reviewer to Section 4.3.1, where we report the runtime of the entire pipeline.
> 3. We clarify that these weights are different since the PSN is trained separately for prediction and planning tasks with different loss features. In designing the loss weight, we tested multiple weight combinations and finalized the current one to best represent the trade-off between time-efficiency and trajectory performance.
> 4. We clarify that in scenarios where the agent number is as large as 20. The result is not sensitive to the choice of the pre-filtering. The only usage of the pre-filtering is to regulate the input size of the network. We have added a discussion in Remark 2.
> 5. We clarify that, given the fast implementation of GIN, it can be called at every step or only at the beginning. In the experiment section, it is only called once for each scenario.
> 6. Yes, the reviewer is right.
> 7. The star symbol is the goal position for each agent; we have updated the figure to make it clear.
> 8. We refer the reviewer to the tables in the Appendix, where we clearly compare our method with the baselines in these metrics.
>
> References:
>
> [1] M. Chahine, et al., “Local non-cooperative games with principled player selection for scalable motion planning,”
>
> [2] T. Qiu and D. Fridovich-Keil, “Inferring occluded agent behavior in dynamic games from noise corrupted observations,”
>
> [3] S. Shi, et al, “Motion transformer with global intention localization and local movement refinement,”
>
> [4] A. Alahi, et al, “Social lstm: Human trajectory prediction in crowded spaces,”

---

### Author Response · Authors · 2026-05-22
**Official Comments to the reviewers**

We thank the reviewers for providing their thoughtful and constructive feedback regarding our work. This feedback has allowed us to improve the technical quality of our work and the clarity and organization with which it is communicated.

We have carefully revised our paper based on the comments we received, and we detail our changes and responses to individual comments in the remainder of this document.

Any changes to the main text are highlighted in blue. A full draft of the text, with edits highlighted, is resubmitted.

---

### Decision · Action_Editor_zZ4J · 2026-06-17

**Recommendation:** Accept as is

**Audience:**

Yes

**Audience Explanation:**

Yes, some of the TMLR's audience would be interested in the findings.

**Claims And Evidence:**

Yes

**Claims Explanation:**

Overall, reviewers believe the submission is a complete work with findings worth sharing with the community. There were remaining concerns on the empirical evaluation after the authors' responses. For example, reviewer 7XZv found the key claims, including computational efficiency and trajectory quality, are only partially supported, and the experiments are limited to simplified double-integrator dynamics with at most 20 agents. Reviewer NtnS mentioned that some issues or weaknesses cannot be solved completely via the rebuttal.

Though the AE believes that reviewer 7XZv's feedback is worth exploring as future work, the submission itself is well supported with accurate, convincing and clear evidence, and is ready for publication.

**Resubmission Of Major Revision:**

The authors may consider submitting a major revision at a later time.